# Fluid Biomarkers in HPV and Non-HPV Related Oropharyngeal Carcinomas: From Diagnosis and Monitoring to Prognostication—A Systematic Review

**DOI:** 10.3390/ijms232214336

**Published:** 2022-11-18

**Authors:** Shaun C. Lee, Karina K. C. Leung, Audrey C. Y. Chung, Elysia S. Y. Wong, Katie L. Meehan, Jason Y. K. Chan

**Affiliations:** 1Faculty of Medicine, The Chinese University of Hong Kong, Hong Kong SAR, China; 2Department of Otorhinolaryngology, Head and Neck Surgery, The Chinese University of Hong Kong, Hong Kong SAR, China

**Keywords:** liquid biopsies, fluid biomarkers, oropharyngeal cancer, tongue base, HPV, detection, diagnosis, treatment monitoring, prognosis

## Abstract

Biomarkers are crucial in oncology, from detection and monitoring to guiding management and predicting treatment outcomes. Histological assessment of tissue biopsies is currently the gold standard for oropharyngeal cancers, but is technically demanding, invasive, and expensive. This systematic review aims to review current markers that are detectable in biofluids, which offer promising non-invasive alternatives in oropharyngeal carcinomas (OPCs). A total of 174 clinical trials from the PubMed search engine in the last 5 years were identified and screened by 4 independent reviewers. From these, 38 eligible clinical trials were found and subsequently reviewed. The biomarkers involved, categorized by human papillomavirus (HPV)-status, were further divided according to molecular and cellular levels. Recent trials investigating biomarkers for both HPV-positive and HPV-negative OPCs have approaches from various levels and different biofluids including plasma, oropharyngeal swabs, and oral rinse. Promising candidates have been found to aid in detection, staging, and predicting prognosis, in addition to well-established factors including HPV-status, drinking and smoking status. These studies also emphasize the possibility of enhancing prediction results and increasing statistical significance by multivariate analyses. Liquid biopsies offer promising assistance in enhancing personalized medicine for cancer treatment, from lowering barriers towards early screening, to facilitating de-escalation of treatment. However, further research is needed, and the combination of liquid biopsies with pre-existing methods, including in vivo imaging and invasive techniques such as neck dissections, could also be explored in future trials.

## 1. Introduction

Biomarkers are crucial in oncology, from the detection and monitoring of cancer to guiding management and predicting treatment outcomes. However, histological assessment of tissue biopsies, which is currently the gold standard for oral cancers, is technically demanding, invasive, and expensive. Liquid biopsies, through the analysis of cancer biomarkers in bodily fluids, offer promising non-invasive alternatives to patient care in oral cancers. The aim of this comprehensive literature review was to explore and discuss current markers that are detectable in various biofluids, including plasma, oropharyngeal swabs, and oral rinse.

Specifically, this review will cover recent advancements in the application of fluid biomarkers to cancers of the oropharynx and the tongue base. These two sites are in close proximity, and the terms are often used interchangeably but give rise to notably different pathologies. Unfortunately, this leads to challenges when aggregating and comparing results in the field. The oropharynx includes the soft palate, lateral and posterior walls of the throat, tonsils, and the back third of the tongue, also known as the tongue base [1]. There is a need to address cancers originating in the oropharynx, as death rates for oropharyngeal cancers (OPC) are rising by 2% per year, in contrast to other subsites of the head and neck. It is estimated that 70% of cancers arising in this subsite are caused by the human papillomavirus (HPV), which could be detected from biofluids [2]. The anterior tongue was thus excluded due to the absence of HPV association. With the rising trend, early detection is of utmost importance, as it can improve 5-year survival rates up to 85% [3]. Unfortunately, almost half of the total oral and OPCs are not detected until spread to adjacent tissues and/or regional lymph nodes has occurred, resulting in a 5-year survival rate of 67%.

## 2. Method and Analysis

We performed a scoping review guided by the Preferred Reporting Items for Systematic Reviews and Meta-Analyses 2020 checklist. A systematic search was performed in PubMed, with 174 articles identified. Search results were limited to the recent 5 years. They were screened by 4 independent reviewers, with 7 being excluded due to duplication. No filters were used, and clinical trials were manually selected. Inclusion criteria included: (1) the tumor is of oropharyngeal origin; (2) fluid biomarkers were investigated; (3) data derived from clinical studies; (4) were conducted in the recent 5 years from 1 Jan 2017 to 1 May 2022. The selection process is summarised in Figure 1.

Search terms included: (oropharyn*) and (carcinoma or cancer or tumo *) and (squamous) and ((circulating tumo * dna) or (circulating tumo * cell) or (cell free dna) or (plasma biomarker) or (saliva biomarker) or (liquid biopsy) or (extracellular vesicle) or (exosome *) or (oral rinse) or (biofluid)) and (clinical).

After the initial search and deduplication, 122 records were later excluded due to irrelevance, and only clinical trials on fluid biomarkers were kept. Titles/abstracts deemed potentially eligible for inclusion were advanced to full-text screening (*n* = 45). Finally, 38 were included in this review. The biomarkers being considered in this review were further divided into categories including HPV-associated, extracellular vesicles (EV), and methylated genes. A significant number of studies on tissue markers were observed. However, since the focus of this research is on biofluid markers, these were not reviewed in detail and will only be referred to when appropriate. Figure 2 summarises the identified fluid biomarkers, grouped according to their type of biofluid and HPV-status.

## 3. HPV-Positive OPC

Oropharyngeal squamous cell carcinoma is classified by the World Health Organization into HPV-positive and HPV-negative types [4], due to their significant differences in epidemiology, clinical features, histology, and prognosis. HPV-positive OPC patients generally show better prognosis and survival compared to their HPV-negative counterparts.

Regarding HPV-positive OPCs, biomarkers are heavily centered around detection of HPV DNA and their associated oncoproteins, aiming to improve outcome and survival. Twenty clinical studies were identified from the systematic search and are categorized into genetics, epigenetics, extracellular vesicles, and oncoproteins.

### 3.1. Genetics

#### 3.1.1. DNA

Within HPV-positive oropharyngeal squamous cell carcinoma (OPSCC), the use of HPV circulating tumor DNA (ctDNA) has been a promising biomarker in various aspects of clinical management, especially in early detection, prognosis prediction, and treatment outcome monitoring. The articles and biomarkers identified have been summarised in Table 1.

##### Detection

Early detection has been associated with lower incidence and better survival. The use of ctDNA or cell free DNA (cfDNA) as biomarkers for early detection has already been established in various types of cancers, such as Epstein–Barr virus (EBV)-associated nasopharyngeal cancers [15]. However, despite the high association of HPV with OPC, many patients tend to present at an advanced stage as their disease is difficult to detect and generally asymptomatic. Nevertheless, clinical early detection may be similarly possible with liquid biopsies.

The potential use of HPV DNA in oral rinse as an early detection tool for OPC has been showcased in two studies. Twelve participants from a cohort of 660 cancer-free individuals were discovered to have positive oral HPV16-DNA and were followed-up for 24 months in a prospective study from Australia [6]. Among the 12, three cases had persistent oral HPV16 infection, with one subsequently developing a stage I OPC while another had mild tonsillar dysplastic lesion. This shows that a high persistent HPV viral load serves as a predictor for chronic infection and HPV-driven cancers including OPSCC. Despite the small sample size, this suggests the potential that a high persistent HPV load may serve as a predictor for a chronic infection and HPV-driven cancers including OPSCC, hence the possible utilization of HPV DNA to screen asymptomatic individuals for OPSCC. A similar conclusion could be drawn from another larger-scale retrospective study which validated that HPV tests on oral rinse could detect OPCs with a positive-predictive value of 94% and sensitivity of 78% [5].

Apart from oral rinse, HPV cfDNA in plasma could also serve the function of detecting OPC. HPV cfDNA could be detected in OPSCC patients with 100% specificity and 72% sensitivity, respectively [7]. Further supporting the rationale that cfDNA in plasma could be used as a diagnostic tool, viral loads in plasma were found to have a positive correlation with that in tumor tissue. However, probably due to the lack of tumor lysis and HPV DNA in the systemic circulation, cfDNA could not be detected in plasma of patients without nodal metastasis, which limits its use in early diagnosis. It was reported in another study that HPV cfDNA E6 and E7 in plasma could be used in combination to detect HPV-positive OPSCC with a specificity and sensitivity of 100% and 77%, respectively [13].

A meta-analysis reviewing 10 studies involving a total of 457 HPV-positive HNSCC patients ascertained the above conclusions. It was reported that HPV cfDNA in blood has diagnostic ability with a pooled sensitivity and specificity of 0.65 and 0.99, respectively [8]. Given that it has a pooled diagnostic odd ratio of 371.66, HPV cfDNA shows sufficient diagnostic accuracy for HNSCC. However, it was noted that the area under curve (AUC) of receiver operating characteristic (ROC) curve for the diagnostic ability of HPV cfDNA was 0.77, which suggests a certain degree of heterogeneity across results.

##### Predicting Prognosis

Apart from detection, liquid biopsy of HPV ctDNA in plasma could be applied to predict prognosis in HPV-positive OPC patients, particularly in the selection of patients who may benefit from de-intensified treatment.

HPV cfDNA E6 and E7 concentrations were found to increase with size of tumor in HPV-positive OPSCC, with the median copy number of E6 and E7 cfDNA per mL plasma being significantly higher in large tumors [13]. Similar results were obtained by Veyer et al., demonstrating that baseline HPV16 ctDNA loads were positively correlated with T, N, and M (tumor, nodes, metastasis) status in HPV-positive OPC patients [10]. Furthermore, median concentrations of HPV ctDNA increases from 1.45 to 4.8 log cp/mL from Stages I to IV, respectively, showing positive correlation with the currently used 2018 AJCC (American Joint Committee on Cancer) staging. Undetectable baseline HPV16 ctDNA is associated with lower staging, with 74% of patients presenting with stage I OPSCC. Positive trends could also be observed between baseline HPV16 ctDNA detection among HPV-positive OPSCC patients and progression-free survival (PFS), as well as mortality rates. This highlights the possibility of using HPV ctDNA as an additional tool for risk stratification apart from histological and clinical diagnosis.

The above findings are partially consistent with those from another study which ascertained the positive correlation of baseline HPV16 ctDNA levels with tumor burden among HPV-positive OPC patients [11]. However, this study further elicited that despite low pre-therapeutic levels of HPV16 ctDNA (≤200 copies/mL) being associated with low tumor burden, it is paradoxically indicative of worse prognosis. Through comparing HPV ctDNA load in plasma with HPV copy number and level of HPV integration in the biopsied tumor genome, it was found that those with lower baseline HPV ctDNA concentrations have lower HPV copy number and higher HPV integration in tumors, which were adverse tumor genomic factors leading to poorer prognosis. This showcases the possible use of HPV ctDNA in prognostication of OPC patients.

There is yet to be a broad consensus on the role of baseline HPV ctDNA for risk stratification. Easily obtainable biofluid including saliva and blood shows the potential value of providing supplementary information for treatment choice, though its clinical application as a prognostic marker awaits large-scale multicentric study to further investigate.

##### Treatment Monitoring

Apart from initial staging, it has been reported that HPV ctDNA concentrations could be used to predict and monitor treatment response by reflecting the dynamic tumor burden.

From a small-scale study, among six patients with post-treatment serum, a patient with negative baseline HPV ctDNA detection and another four that had significantly decreased HPV ctDNA loads post-treatment showed a complete clinical response to treatment, whereas the patient having fivefold increase in HPV ctDNA load after treatment died within days [10].

A similar conclusion could be drawn from research conducted by Haring et al. in which changes in plasma HPV16 ctDNA concentrations were shown to correlate with radiographically determined treatment responses for OPC patients on immunotherapy [9]. Results suggested that patients with a <60% increase in HPV16 ctDNA is associated with a favorable response without disease progression, whereas those with a ≥60% increase had disease progression despite therapy. Furthermore, changes in HPV16 ctDNA concentrations were found to precede determination of radiographic response, even up to 100 days in particular cases. HPV16 ctDNA could also effectively identify those who are having radiographic pseudo-progression, with its load decreasing in accordance with the molecular response. Furthermore, HPV ctDNA could also be used to predict treatment response for advanced HPV-positive OPSCC patients undergoing chemoradiotherapy (CRT). All patients that had complete radiological response demonstrated HPV DNA levels below the threshold, which was defined as 10 reads in 7 amplicons [12], reiterating its function as a dynamic marker for therapeutic response.

With more accurate predictions on treatment response, patients could be selected for de-escalation of treatment and avoid unnecessary biopsies and surgical treatment.

##### Post Treatment Surveillance

Recurrences of OPSCC mostly occur within the first two years following treatment completion. HPV-positive OPSCC can recur up to five years or more after therapy [15,16]. It is crucial to have a readily obtainable biomarker to monitor conditions of patients regularly over a long period of time.

In a prospective clinical trial, HPV ctDNA in plasma demonstrated the ability to identify disease recurrence in non-metastatic HPV-positive OPSCC patients receiving CRT. With a median follow-up time of 23 months post-treatment, all 87 patients that had undetectable HPV ctDNA throughout did not develop recurrence [14]. Among the 28 patients showing positive HPV ctDNA during post-therapeutic surveillance, in which 16 of them had two consecutive plasma samples with detectable HPV DNA, 15 patients were diagnosed with biopsy-proven recurrence. This shows that recurrence could be reflected by two consecutively positive HPV ctDNA blood tests with a positive predictive value (PPV) of 94%, indicating serial HPV measurements may be useful in risk stratification with higher accuracy. Similar to its application in prognosis, detection of post-treatment HPV ctDNA positivity was also found to precede biopsy-proven recurrence by a median of 3.9 months.

This is also illustrated by Lee et al. where a patient in the study showed elevated HPV DNA level despite lack of active disease at the primary site or cervical lymph nodes [12]. Positron emission tomography (PET) showed increased 18F-fluorodeoxyglucose (18F-FDG) uptake from the liver, however, HPV DNA load demonstrated marginal reduction after resection of the liver and paradoxically started to increase continuously. PET subsequently revealed increased uptake in cervical lymph nodes 8 months after the surgical resection of liver, confirming the ability of plasma HPV DNA to detect recurrence prior to radiographic imaging. Another study specifically looking into the use of HPV cfDNA E6 and E7 ascertained the above findings. 5 patients (67.5%) with HPV-positive OPC showing two consecutive positive HPV loads after treatment subsequently developed recurrence, and HPV cfDNA was observed to increase prior to that diagnosed by routine examination [13].

#### 3.1.2. RNA

Two articles have been identified regarding RNA biomarkers for HPV-positive OPC, and their biomarkers are summarised in Table 2.

##### Detection

The use of genetic markers for the detection of HPV-positive OPC extends to RNA apart from DNA. Presence of HPV E6/E7 mRNA indicates transcriptionally active HPV, which is involved in the carcinogenesis of OPSCC. HPV E6/E7 mRNA from fine needle aspiration (FNA) of neck lymph nodes of metastatic OPSCC patients showed an overall agreement of 88%, which is comparable to conventional immunohistochemical p16 staining of tumor tissue, allowing it to serve as a tool for diagnosis [16]. FNA samples were found to have higher accuracy as compared to oral samples.

##### Prognosis

Instead of focusing on HPV, a prospective study observed the role of oral and gut microbiomes on HPV-positive OPSCC patients after CRT through measuring rRNA of bacteria [17]. Pre-therapeutic oral and gut microbiomes were found to differ among different stages of HPV-positive OPC. For oral microbiome there was a significant enrichment in four genera (Fusobacterium, Gemella, Leptotrichia and Selenomonas) in stage III patients compared to stage I-II patients, whereas gut microbiome showed significant enrichment in two phyla: Actinobacteria and Proteobacteria in stage III patients. Further trials are required to research into whether such differences in baseline microbiome composition could be used for risk stratification in OPC patients.

##### Treatment Monitoring

Composition of microbiome in oropharyngeal swabs was revealed to correlate with the treatment status, with the number of species in oropharyngeal swabs decreasing significantly after therapy. A shift in taxonomic composition could also be observed with an increase in relative abundance of gut-associated obligate anaerobes, such as Bacteroides species. However, the underlying reason behind such a change is yet to be discovered.

With emerging interests on the action on microbiome, the possible implications of how novel development in the field could aid in the diagnosis and monitoring of OPC is worth further exploration.

### 3.2. Epigenetics

Findings of the two studies looking into the use of epigenetic biomarkers are summarised in Table 3.

Studies have investigated a range of novel methylation markers aiming to identify suitable ones for diagnostic or monitoring purposes. Various genes were found to associate with OPC, warranting studies to assess the relationship between epigenetic modifications and clinical progression of the cancer.

Calmodulin-like 5 (CALML5) is a skin-specific calcium binding protein which is involved in regulation of epidermal differentiation [20]. Translocation of Yes-associated protein 1 (YAP1), an oncogenic driver of HNSCC [21], into the cytoplasm is hindered by methylation of CALML5, resulting in increased transcription in carcinogenesis [22,23]. As a membrane-bounded protein with a glycosylphosphatidylinositol anchor, lymphocyte antigen 6 complex locus D (LY6D) has an established role in the adhesion of head and neck cancer cells to endothelial cells in HNSCC patients [24]. Hypermethylation of the gene promoter region was also reported in lung cancer [25]. It was illustrated in a verification study that the methylation status of CALML5 and LY6D was associated with reduced survival with a hazard ratio of 7.01 and 10.69, respectively [18].

DNAJ heat shock protein family member C5 gamma (DNAJC5G) belongs to the DNAJ family, in which various members were revealed to have different interactions with viruses including adenovirus, vaccinia virus, HIV-1 [26], and HPV16 E7 oncoprotein [27]. Hypermethylation of CALML5, LY6D, and DNAJC5G were demonstrated to correlate with clinical condition of HPV-positive OPSCC patients, with the number of patients showing methylation of the genes significantly reduced post-treatment as compared to their pre-treatment statuses [18].

Post-translational modification of genes in the salivary rinse has also been investigated to identify markers for post-treatment surveillance of HPV-positive OPSCC. It has been shown that a combined panel of HPV DNA (HR E5L2-4) and hypermethylation of endothelin receptor type B gene (EDNRB) gave the best sensitivity (0.90) and specificity (0.81) in detecting recurrence [19]. The panel was also able detect recurrence earlier than clinical detection by 2.4 ± 1.6 months. This adds on to previous studies which established hypermethylation of EDNRB with the presence of invasive OPSCC and increased risk of locoregional recurrence [28,29].

With promising results, these liquid biopsy markers may be utilized complementary to conventional radiographic examination in long-term patient surveillance.

### 3.3. Extracellular Vesicles (EVs)

Extracellular vesicles are actively released by cells to facilitate cell-to-cell communication. They contain molecular cargo such as nucleic acids and proteins, and are present in all biological fluids, making EVs valuable and worthy of investigation. It has been established that in HPV-related cervical cancer, their contents, in particular microRNAs, are altered. Two articles have been identified and their corresponding biomarkers are summarised in Table 4.

#### 3.3.1. Detection

In an in vitro study on oropharyngeal tumor cells, it was revealed that HPV-positive OPSCC cells produced less EVs than HPV-negative cells. Their cargo was analyzed and revealed an abundant number of miRNAs independent of HPV status, with 9 common miRNAs found to be present in all samples. Pathway analysis showed that PI3K-Akt, Fox0, HIF-1, mTOR, and p53 signaling pathways were targeted by these miRNAs. However, 14 and 19 miRNAs were identified to be only enriched in HPV-positive and HPV-negative groups, respectively. miR-99a-5p, miR-27a-3p, and miR-27b-3p, which were previously reported as cargo of EVs from HPV18 + HeLa cervical carcinoma cells, were also found to be enriched in HPV positive OPSCC-produced EVs. Expression of miR-20b-5p, which is also enriched in HPV-positive OPSCC-derived EVs and promotes cancer cell invasion, has been proven to be dependent on the viral E6 oncoprotein. Therefore, although EVs of different origins may have overlapping miRNA profiles, certain miRNAs are dysregulated in specific subtypes, providing potential for detection of OPSCC [30].

#### 3.3.2. Recurrence

One major challenge to developing clinically significant miRNA biomarker panels is the biological variability and differences in detection methodology, including collection, storage, RNA isolation, and processing, thus leading to unreproducible findings. In a study of 40 OPSCC patients, a cross-validated StaVarSel (stable variation selection) method was utilized and produced a 11-miRNA-ratio model that accurately identified HPV-positive OPSCC (90% sensitivity and 79% specificity), showing good potential for diagnosis and predicting recurrence. However, the subjects were in advanced stages of OPSCC and thus the ability of EV signatures in early detection was not tested [31].

### 3.4. Oncoproteins

Three articles investigating the use of oncoprotein biomarkers in OPC have been identified and summarised in Table 5.

#### 3.4.1. Early Detection and Classification

Patients with HPV + OPSCC have better overall and disease-free survival. Determination of HPV status is therefore important for risk stratification. However, the surrogate immunohistochemical marker, p16(INK-4a), which is used clinically with high sensitivity, only has a specificity of 83%, leading to potential misclassification of a specific subgroup of patients who are p16(INK-4a) positive but HPV DNA-negative. These patients have been reported to have distinct clinical features, worse prognosis, and increased metastasis potential compared to both p16(INK-4a) and HPV DNA-positive patients.

Thirty-six patients with newly diagnosed OPSCC were analyzed with three serum markers, which were increased in the sera of HPV-associated OPSCCs, and were identified to differentiate HPV-associated and HPV-independent OPSCCs, including ApoF, Galactin-3-Binding protein, and complement component C7 [32]. ApoF is a lipid transfer inhibitor protein that regulates cholesterol transport. It is hypothesized to be increased due to metabolic reprogramming by malignant cells or stromal reaction by HPV per se. Galactin-3-Binding protein, a glycoprotein implicated in self-nonself discrimination, is upregulated in viral infection, with higher levels being linked to poor prognosis and progression in cancers not limited to OPSCC. Complement C7 is an important component of the membrane attack complex (MAC), and can activate oncogenic MAPK, ERK, PI3K, and Ras pathways, which are important in the progression of HPV-infected cells toward malignancy. Serum proteomes therefore have potential in providing simple and cost-effective ways to assess HPV status and assist classification of patients.

Oral testing is another focus of non-invasive biomarker development, due to the close proximity to oropharyngeal tumors in addition to the easy collection of saliva and swabs. A study compared HPV E6 oncoproteins and E6/E7 mRNA in fine-needle aspiration samples and oropharyngeal samples (saliva and oral swabs) and found that HPV E6 oncoprotein tested in such samples agreed with results of p16 or HPV tested tumors, but at a lower concentration and rate than mRNA and DNA [16]. Oral testing for oncoproteins is currently not as informative as DNA and RNA counterparts towards detecting malignancies, and further studies and development are required for sample optimization.

#### 3.4.2. Predicting Prognosis

Although HPV-positive OPSCC has a more favourable prognosis, 10–25% still recur. A small single-institution study evaluated the possibility of using a liquid biopsy as a prognostic marker, measuring HPV E6E7 expression in circulating tumor cells (CTCs). Pre-treatment baseline expression of HPV E6/E7 oncogene in CTCs were associated with both progression-free survival and overall survival, with patients having significantly higher risk of relapse and death. However, the presence of E6E7 + CTCs at the end of treatment was not significant in PFS and overall OS [33]. Research is still undergoing to combine the biomarker with AJCC staging data to improve risk definition and discrimination.

### 3.5. Immune Response

A range of immune response-related markers have been identified to have prognostic and surveillance functions, which are summarised in Table 6.

#### Predicting Prognosis

Immunological responses to HPV may provide insights for treatment response as well. A correlation is likely between the HPV-positive status of the tumor and T cell response to HPV antigens, as demonstrated by a study which found that 80% of patients who had detectable HPV16 DNA in tumors also had CD4+ or CD8 + T cell response to HPV16 E6/E7. This study provided survival data in OPSCC stratified by cell-mediated immune response to HPV16 peptides, and demonstrated that enhanced immunoreactivity to antigen E7 was correlated with longer disease-free survival. The average disease-free survival (DFS) for all patients was 43.7 months, and was increased to 47.3 months for the HPV+ cohort. Higher average DFS (49.6 months) was observed in patients with an increased CD8 + T cell response to E7. Cox regression analysis revealed that a retained or enhanced CD8 + T cell response to E7 has significant influences on DFS [34]. Rising CD4 +/CD8 + T cell response may also indicate recurrence of disease, but requires further studies looking at a more distant time-point after treatment.

Apart from T cell response, antibodies directed against HPV antigens have also been investigated as prognostic biomarkers pre- and post-treatment. With removal of the tumor and thus the source of antigens, B cell responses should in theory become limited. A study on 77 patients revealed better survival outcomes for patients with higher levels of anti-E2 IgG. Interestingly however, the presence of IgA antibodies does not appear to be linked to survival outcomes. It was also found that levels of anti-E6 and E7 IgG antibodies differ pre- and post-treatment. However, their application on predicting recurrence is still unknown due to limitations of the cohort.

## 4. HPV-Negative OPC

Although some oropharyngeal cancers are HPV-negative, there are still alternative biomarkers that could assist in detection and management. Eighteen clinical studies were found in the systematic search and are categorized into genetics, protein, and immune response-related biomarkers below.

### 4.1. Genetics

Three articles and biomarkers for HPV-negative OPC have been summarised in Table 7.

#### 4.1.1. Early Detection

DNA methylation patterns of genes used as biomarkers may serve the early diagnosis of OPSCC. A previous study conducted in Thailand identified that methylation at cg01009664 of the thyrotropin-releasing hormone (TRH) gene may be a potential marker for OPSCC screening by a bioinformatics approach [36]. Within the validation cohort of 24 OPSCC patients, average TRH methylation levels of oral rinse samples from OPSCC patients (3.54 ± 0.37 ng/μL) was significantly higher than that of healthy controls (2.86 ± 0.64 ng/μL). With a cut-off value of 3.31 ng/μL, there was sensitivity and specificity of 82.61% and 92.59%, respectively.

#### 4.1.2. Predicting Prognosis

Another candidate biomarker is the DEK oncogene, which was shown to play an important part in tumorigenesis as a chromatin structural and remodeling protein. Downregulation of plasma DEK oncogene was found to be associated with poor prognostic factors including HPV-negative status and advanced tumor stage in HNSCC patients [37]. Plasma DEK concentrations in p16 negative HNSCC patients (390.4 pg/ml) was found to be significantly lower than those with p16-positive patients (668.6 pg/ml). Median plasma DEK concentrations of patients with large tumors (T3-4) (401.0 pg/ml) are significantly lower than those with small tumors (T1-2) (675.1 pg/ml).

#### 4.1.3. Treatment Selection

Previous studies have explored the possibility of identifying gene mutations by next-generation sequencing for target therapy. DNA damage repair (DDR) gene mutations in ctDNA within an 18-gene panel were detected in 37.0% of the HNSCC patients, which included 68 (40.0%) OPC patients [38]. Specifically, BRCA2 and ARID1A were found to have the highest prevalence among the cohort. Although the prevalence of such mutations is lower in oropharyngeal primaries across all gene subsets compared to SCC of other anatomical sites (*p* = 0.01), this still showcases the possibility of incorporating poly ADP-ribose polymerase (PARP) inhibitor therapy in the treatment of certain patients, since cells without functional DDR genes for homologous recombination repair (HRR) pathway are sensitive to PARP inhibition. Thus, testing for DDR gene mutations in ctDNA may help to identify patients that are susceptible to target PARP inhibitor therapy, allowing another option aside from the conventional ways of treatment.

### 4.2. Proteins

A range of protein biomarkers were found to have diagnostic or prognostic function for HPV-negative OPC, which is summarised above in Table 8.

#### 4.2.1. Early Detection

Matrix metalloproteinases (MMPs) play a role in the regulation of extracellular matrix (ECM) remodelling, in which MMP-1, also known as collagenase-1, was the first to be discovered in the family in 1962. Being responsible for initiating the degradation of native fibrillar collagens by cleavage of peptide bonds [44], MMP-1 has been reported to be strongly associated with metastasis, angiogenesis, and inflammation in tumorigenesis [45,46]. A recent retrospective study has shown that salivary MMP-1 may serve as a promising diagnostic marker in oral squamous cell carcinoma, including cancer in the oropharynx. Salivary MMP-1 was found to be significantly associated with OPSCC. At the best cut-off value of 122.5 pg/ml, salivary MMP-1 showed sensitivity and specificity of 70.37% and 74.64%, respectively, with an accuracy of 74.51% [41], indicating its potential ability to be used in early-stage screening in high-risk populations.

Diagnostic value was also found in another secretory enzyme lysyl oxidase like 2 (LOXL2), which was speculated to be involved in ECM remodeling and epithelial-mesenchymal transition, thus contributing towards tumor progression [47]. Strong association between serum exosomal LOXL2 levels and low-grade HNSCCs was revealed. The average LOXL2 levels in HNSCC patients was over 9-fold higher than that of the healthy group [40]. Significant differences were detected specifically between stage I/II patients versus healthy patients, enabling LOXL2 to be utilized in early detection of HNSCC.

Neurotrophin-3 (NT-3) is known to be involved in various cancers including breast cancer [48], lung cancer [49], and adenoid cystic cancer [50]. A recent study of 19 patients also showed evidence for its application in OPSCC [39]. Plasma NT-3 levels were found to be downregulated in SCC patients, allowing differentiation between patients and the healthy population.

#### 4.2.2. Predicting Prognosis

Numerous studies have investigatated the possibility of utilizing enzymes in the MMP-1 cascade to predict prognosis in OPC. ProMMP-1, the inactive precursor of MMP-1, gains function via a proteolytic process involving proteases and MMP-3. The activity of MMP-1 is further regulated by the enzyme tissue inhibitor of metalloproteinase-1 (TIMP-1) [51]. TIMP-1 also functions as a growth factor by binding to cell surface ligand CD63, resulting in activation of intracellular signalling via focal adhesion kinase (FAK) which leads to cell proliferation [52].

The prognostic value of MMPs has been demonstrated, with levels correlating with factors involved in cancer staging, including tumor size, nodular metastasis, and degree of cancer cell differentiation. Salivary MMP-1 levels were found to increase significantly between stage I and stage IV patients, thus reflecting cancer progression [41]. Moreover, salivary MMP-1 levels of T4 patients (2294.5 pg/ml) showed significant differences to those of T0-1 patients (429.6 pg/ml). Regarding nodal metastatic status, salivary MMP-1 levels in patients with one (N1) or multiple infiltrated lymph nodes (N2-3) were higher than those without (N0). These results were consistent with the functional role of MMP-1 in ECM remodeling associated with metastasis. As for degree of differentiation, salivary MMP-1 levels in poorly differentiated (G3) cancers show a median difference of 4.5-fold when compared to those with undifferentiated and well-differentiated (G0-1) patients.

Another novel study suggested the use of MMPs in surgical drain fluid instead of conventional saliva and blood samples, since it is obtained routinely and eliminates effects due to proximity from tumor sites, an issue in salivary samples. Measurements of MMP-1 and MMP-3 from fluid samples were collected postoperatively every 8 h [42]. Within the 20 patients with SCC of the oral cavity and oropharynx, patients with recurrence at one year were revealed to have significantly lower levels of MMP-1 and MMP-3, with a relative difference of 2.78 and 5.29, respectively.

Apart from MMP itself, overexpression of TIMP-1 in serum was also associated with poor prognosis specifically in HPV-negative OPC. Carpén et al. demonstrated the concordance of TIMP-1 with OPC among 90 patients, showing that high TIMP-1 serum levels were strongly associated with poorer overall survival (adjusted HR 14.7) and disease-free survival (adjusted HR 8.7) among HPV-negative patients [43]. This serves as a potential prognostic marker for HPV negative oropharyngeal cancers, which is associated with poorer outcome.

### 4.3. Immune Response

Ten clinical studies were identified that investigated the potential of utilizing peripheral changes in immune response as markers for HPV-negative OPC. Blood, saliva, and post-surgical drain biofluids were investigated. The applications of these biomarkers included early detection, tumor staging, as well as assessing prognosis and predicting survival.

The changes in immune response can be categorized into genetic, cellular, and tissue levels, and are summarized in Table 9 below.

#### 4.3.1. Early Detection

Salivary lactate dehydrogenase (LDH) and plasma interleukin-1 receptor antagonist (IL-1Ra) levels were hypothesized to be upregulated in patients with cancers of oropharyngeal origin.

#### 4.3.2. Salivary LDH

Tumor cells increase metabolic processes, including glycolysis, to meet the heavy energy demands. LDH is increased in cancer cells to catalyze pyruvates into lactates. It is also expressed during cellular necrosis and in HNSCC patients has higher levels than that of the healthy control group, as found in a study on 44 HNSCC patients [53]. It is notable that this difference is more markedly increased in the oropharyngeal tumors than in other HNSCCs. However, LDH expression does not correlate with histopathological stage nor grade. Further research is also needed to compare LDH levels before and after the onset of cancer, as well as to monitor the changes during the course of treatment, which could provide much more valuable information and reduce the influence of other covariant factors.

##### Plasma IL-1Ra

Plasma IL-1Ra was also investigated as another potential marker for early detection in a study on 87 patients [54]. Being a member of the interleukin family, it is a major IL-1 antagonist, preventing binding of IL-1 to its receptor and thus possessing anti-inflammatory and innate immunity regulatory actions. It is endogenously produced by immune cells, epithelial cells, and adipocytes, and together with the IL-1 family, affects carcinogenesis and progression. However, pre-diagnostic plasma levels were not significantly increased, and no predictive diagnostic value was found.

#### 4.3.3. Tumor Staging

Cytokines, immunosuppressive cells, and membrane-bound enzymes have been found to have potential in assisting tumor staging in recent studies including IL-1Ra and CD73.

##### IL-1Ra and the Inflammatory State

Although IL-1Ra was found to not have a diagnostic value, its level correlates with tumor size and body mass index (BMI). Larger tumors (T3, T4) had higher IL-1Ra levels than those with T1 and T2 smaller tumors, and this was seen in all sub-locations of HNSCC. However, nodal involvement and metastasis did not correlate with IL-1ra levels. The increase could be a response to the increased inflammation in the tumor micro-environment. This would inhibit anti-tumor immune responses, leading to continuation of carcinogenesis and tumor growth. The explanation by increased inflammation is supported by the correlation with BMI, as obese patients, who were found with higher IL-1Ra levels, had more adipose tissue and thus increased inflammation.

##### CD73 and MDSCs and the Adenosine Pathway

Apart from circulating cytokines, myeloid-derived suppressor cells (MDSCs) in peripheral blood have been found to directly correlate with clinical stages of HNSCC. MDSCs are a heterogenous group of immature myeloid cells, including macrophages, granulocyte progenitors, and dendritic cells. They can be divided into monocytic (M-MDSC) and polymorphonuclear (PMN-MDSC) forms. PMN-MDSC levels rise in cancer patients, leading to T-cell immunosuppression through the adenosine pathway. A small study on 53 patients revealed that these cells can be indirectly identified and measured through the enzyme CD73 (ecto-5′-nucleotidase), which is expressed on cell surfaces and contributes towards MDSC-mediated immunosuppression [58].

Extracellular adenosine suppresses effector T-cells and stimulates angiogenesis through activation of the adenosine receptors A2A and A2B. Precursors are dephosphorylated into adenosine by CD73, which is found, membrane-bound, on multiple cell types, including tumor cells and tumor infiltrating lymphocytes [62]. Both PMN-MDSC and CD73+PMN-MDSC levels correlate with clinical TNM staging. M-MDSC and PMN-MDSC levels, although both increased in HNSCC patients, were independent of each other, and M-MDSCs had no significant correlation with tumor staging.

It is important to note the small study size which led to large variation in the samples. Tumor differentiation was also not explored in the study. Considering how CD39 plays an important step in hydrolyzation before CD73-mediated dephosphorylation in the adenosine pathway, it also has potential for further investigation. Given the significant impact of immunosuppression by MDSCs through the adenosine pathway, ectonucleotidase inhibitors are promising pharmacological treatments in immunotherapy against HNSCC.

##### Peripheral CTLA-4^+^ and PD-1^+^ Lymphocytes

Immune checkpoints have been a hot topic in the recent decade, with CTLA-4 and PD-1 being major targets in immunotherapy. PD-1 is a transmembrane co-inhibitory receptor of the IgG family, expressed upon activation of T and B cells on many immune cells, including monocytes, natural-killer cells, dendritic cells, T-cells, B-cells, and tumor-infiltrating lymphocytes. A total of 26 HNSCC patients were investigated with flow-cytometric analysis, and results showed that although inhibitory T-lymphocytes preferentially concentrate at the tumor sites, elevation of peripheral CTLA-4+ and PD-1+ cytotoxic T-lymphocytes was still found in patients with stage IV tumors [55]. It is notable that levels were significantly higher (10-fold) in patients negative for HPV than in those positive with HPV. These immune checkpoints thus, apart from guiding management, have potential in assisting in tumor staging, especially for HPV-negative HNSCC cancers including those of oropharyngeal origin.

Another study investigated tissue-resident memory T cells (TRM) in the tumor microenvironment and peripheral TRM-like cells in circulation, which also highly express PD-1 and TIM-3. The specific roles of TRM cells have not been established, but they have direct cytotoxic and antitumor effects through producing effector cytokines. It has been found that TRM-enriched tumors correlate with HPV status, proportion of oropharyngeal lesions in HNSCC patients, and favorable prognosis. CD69+ PD-1+ TIM-3+ TRM-like cells, which circulate peripherally in blood, were also found in the tumor micro-environment. As it is often difficult to collect tumor samples from metastatic sites of HNSCC, the possibility of measuring peripheral PD-1+ TIM-3+ TRM-like cells were investigated. However, blood from 60 HNSCC patients, 50% positive with HPV, were analyzed and no significant correlation was found between peripheral TRM-like cells and clinical features of HNSCC patients [56]. Given the small sample size and the lack of characterization of circulating TRM-like cells, further research is needed.

PD-1 interacts with two ligands, PD-L1 and PD-L2, which are expressed widely in non-lymphoid tissues, including cancer cells. Although circulating PD-1 positive cells have potential applications in tumor staging and management, a study measuring free serum PD-L1 protein levels in 30 patients with an electrochemiluminescence assay found no correlation with PD-L1 levels in tumor and surrounding immune cells of 101 patients with oral tongue squamous cell carcinoma (OTSCC) [57]. There was also no significant correlation with clinicopathological features. Given the importance of PD-1/PD-L1 binding in regulating trafficking, migration, and even eviction of T-cells from the TME, it is interesting how circulating PD-1+ T-lymphocytes correlate with clinicopathological features and clinical outcomes of tumors while its PD-L1 counterparts in peripheral blood do not. It must be noted that tumor cell PD-L1 levels in solid biopsies do correlate with tumor-infiltrating lymphocytes levels and prognosis.

#### 4.3.4. Predicting Prognosis

##### Genetic Level–HLA Traits

Smoking and alcohol consumption are major risk factors for OPC aside from HPV status. However, genetic variations have been proven to play a role in altering the risk of these lifestyle factors in individuals, as well as contributing independently to the risk of carcinogenesis.

A study on 90 patients with HNSCC (31.1% being oropharyngeal in origin) with varying TNM stages conducted HLA typing on leukocytic DNA samples extracted from peripheral blood [60]. Patients overall were observed to have altered HLA frequencies, particularly HLA-B*13, B*44, B*52, B*57, Cw, and DRB4. No significant association between HLA frequencies and HPV status (HPV16 DNA and RNA) was detected, but particular HLA traits were associated with shifts in distribution in sex, tobacco, and alcohol consumption. Carrying any HLA homozygosis was also found to impair progression-free survival.

Well-established risk factors of sex, age, location, HPV-status, and TNM stages were analyzed beforehand and had no significant impact on PFS on their own apart from T4 stage HNSCC. However, when HLA traits were added as a covariate to the model, the aforementioned factors were significantly associated with PFS. HLA-A, B, and Cw play a vital role in presenting aberrant peptides from mutated proteins to CD8^+^ cytotoxic T lymphocytes, leading to destruction of cancerous cells. It is suggested that altered frequencies affect HLA ability in binding and presenting tumor associated antigen peptides, thus impairing immunosurveillance.

##### Cytokine sFLT-1 in Post-Surgical Drain Fluid

Saliva and blood are not the only biomarkers that can be used in OPCs. Post-surgical drain fluid can also provide markers that potentially represent both the tumor condition and also the stages of wound healing. The normal phases of post-operational wound healing, including haemostatic, inflammatory, and angiogenic stages, are tightly regulated by cascades of cytokines, chemokines, and proteases. These levels are expected to change with time over the course of healing and could be monitored by fluid collection at regular intervals after surgery.

It is known that pro-angiogenic factors and anti-angiogenic factors are associated with tumor prognosis. Vascular endothelial growth factors (VEGF) and soluble fms-like tyrosine kinase 1 (sFlt-1), which is an antagonist of VEGF-1, were studied in a small study of 20 patients. Low levels of sFlt-1 in post-surgical fluid were found to have significant correlation with tumor recurrence, while VEGF-A levels were associated with nodal metastasis [42].

Post-surgical drain fluid is not just limited to measurements of cytokines; proteins and enzymes such as metalloproteases have also been found to correlate with tumor prognosis and recurrence. It is therefore a promising method of non-invasively monitoring progress following tumor resection.

##### Cellular Level–Neutrophil-Lymphocyte Ratio

Cells which inhibit tumor growth have been investigated in multiple studies. Mononuclear cells (MNCs), which also reduce cancer proliferation, were investigated using in vivo and in vitro zebrafish models involving OTSCC cells. However, peripheral circulating MNCs were found to have no significant effect on OTSCC cell migration [61].

Pre-treatment leukocytic count was also investigated, but in p16-positive OPSCC patients. Neutrophil lymphocyte ratio (NLR), a marker of inflammation, was measured prior to chemoradiation therapy in 167 patients [59]. Tumor stage correlates with NLR and was thus treated as a covariant. A cut-off of 5 was chosen to categorize patients with high and low states of systemic inflammation, according to a previous systematic review [63]. The median NLR in the cohort was 2.9, and multivariate analysis showed association between NLR >5 with decreased overall survival (OS) and PFS. The rate of recurrence was much higher in patients with NLR >5 (30.8%) than those with low systemic inflammation (8.4%). This could be explained by the different roles of neutrophils and lymphocytes. Neutrophil infiltration enhances tumor growth and progression through extracellular matrix remodeling and enhancing inflammation, while lymphocytic infiltration controls cancer progression through cell-mediated immune mechanisms. Other previous studies on non-HPV associated HNSCC also provided evidence supporting the data of this multivariate analysis [64,65,66].

##### Tissue Level–Anemia

The same study on p16-positive OPSCC patients found hemoglobin concentration, another hematological parameter, to correlate with OS and PFS. Recurrence states however, although higher in patients with pre-treatment anemia (23.1% vs 9.1%), did not have a statistically significant increase. Anemia reflects tumor oxygenation, an important factor affecting the tumor micro-environment. Worse prognosis in anemic patients has been well-established in previous studies involving head and neck tumors. A possible explanation is that pre-existing hypoxic conditions in the tumor micro-environment are exacerbated due to the decreased oxygen-carrying capacity of red cells. Resulting effects include a dampened response of cancer cells to cytotoxic therapy. Further investigation is needed to determine if anemia can be a marker to predict treatment response.

## 5. Limitations

There were several limitations in this systematic review. Firstly, no statistical nor sensitivity analysis was done to assess the robustness of the synthesized results. Furthermore, reporting bias and certainty assessments were lacking. During the review of eligible papers, it was also observed that many trials studied HNSCCs as a whole, with OPCs being only a proportion of the test. Some of the data regarding the distribution of cancer types were also missing and could lead to decreased reliability of data when focusing on OPC alone.

## 6. Conclusions and Future Perspectives

Recent clinical trials investigating biomarkers for both HPV-positive and HPV-negative cancers have approaches from various levels and different biofluids. Promising candidates have been found that could aid in detection, staging, and predicting prognosis of OPCs, in addition to well-established factors including HPV, alcohol consumption, and smoking status. These studies also emphasize the possibility of enhancing prediction results and increasing statistical significance by multi-variate analyses.

Liquid biopsies, being non-invasive methods, offer promising assistance in enhancing personalized medicine in treating cancer patients. The majority of fluid biomarker research is still in plasma, with HPV ctDNA remaining the most-studied fluid biomarker, which has already been applied clinically to other HPV-related cancers. On the other hand, non-HPV related OPCs often present at younger ages, and oral rinses may lower perceived barriers and encourage screening due to its non-invasiveness and simplicity. Markers such as HLA traits may also serve a role in enhancing prognosis prediction when added as covariates for well-known factors including smoking and alcohol consumption status. Furthermore, de-escalation of treatment is potentially improved with the introduction of reliable fluid biomarkers, as they can reduce the need for unnecessary invasive biopsies and monitoring, and even being able to detect recurrence before routine radiographic imaging.

However, further research is still needed, with larger sample sizes and more specific categorization of marker subtypes. In addition, a combination of these markers with pre-existing methods, including in vivo imaging techniques such as PET scans, and invasive techniques such as neck dissections, could also be explored in future trials. With better stratification of patients for HPV-positive cancers, there are also potential implications for treatment deintensification to reduce treatment complications yet sustain survival for these patients.

## Figures and Tables

**Figure 1 ijms-23-14336-f001:**
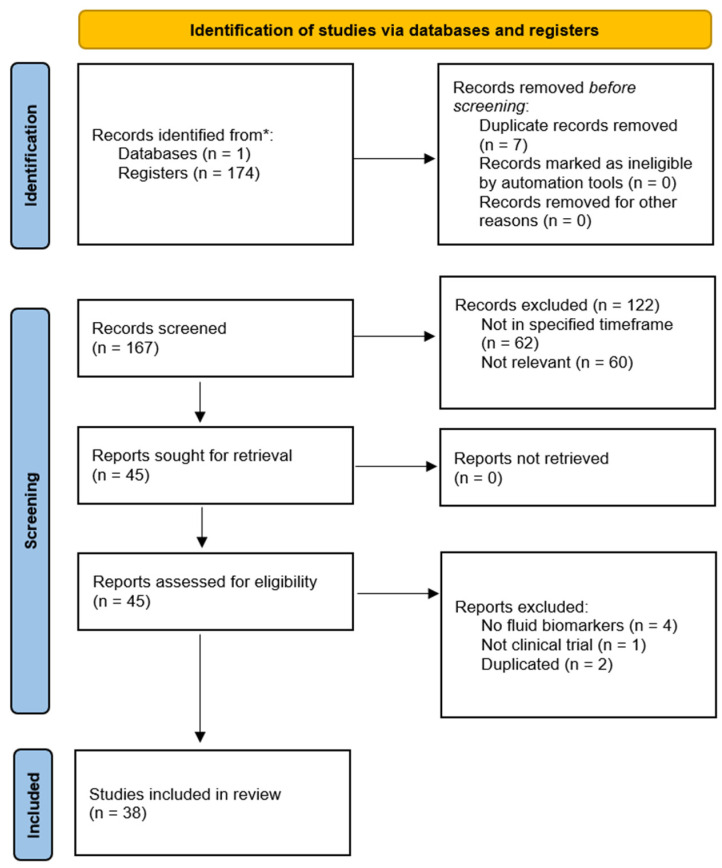
PRISMA flow diagram depicting the selection process articles and reports in the current scoping review.

**Figure 2 ijms-23-14336-f002:**
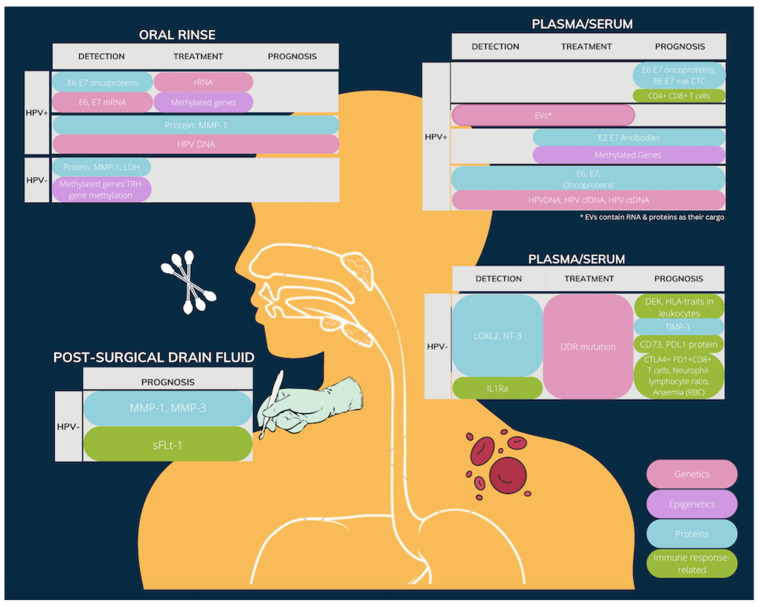
A summary of the identified fluid biomarkers, grouped by location & HPV-status.

**Table 1 ijms-23-14336-t001:** Summary of DNA biomarkers for HPV-positive OPC.

Study	Biomarker	Biofluid	Application	Study Type	Sensitivity	Specificity	Number of Patients (M/F)	Age Range	Mean Age	Other Recorded Covariates
Rosenthal et al. [5]. 2017	HPV DNA	Oral rinse	Detection	Retrospective	0.78	0.50	38:7	/ *	/	Tobacco, alcohol, tumor stage, treatment received
Tang et al. [6]. 2020	HPV DNA	Oral rinse	Detection	Retrospective	/	/	280:370	18–89	52	Tobacco, alcohol, race
Mazurek et al. [7]. 2019	HPV cfDNA	Plasma	Detection	Cross-sectional	1.00	0.72	41:222	30–79	61	/
Campo et al. [8]. 2021	HPV cfDNA	Plasma	Detection	Meta-analysis	0.65	0.99	457	/	/	Tobacco, alcohol, tumor stage, treatment received
Haring et al. [9]. 2021	HPV ctDNA	Plasma	Prognosis, Treatment monitoring	Prospective	0.889	0.889	9:3	/	62	Tobacco, tumor stage, treatment
Veyer et al. [10]. 2020	HPV ctDNA	Plasma	Prognosis, Treatment monitoring	Prospective	/	/	46:20	43–92	65	Tobacco
Chera et al. [11]. 2019	HPV ctDNA	Plasma	Prognosis, Treatment monitoring	Prospective	0.89	0.97	92:11	/	60	Tobacco
Lee et al. [12]. 2017	HPV ctDNA	Plasma	Post-treatment surveillance	Prospective	1.00 (test), 0.90 (validation)	0.93 (test), 1.00 (validation)	55	/	/	Tobacco
Reder et al. [13]. 2020	HPV cfDNA, E6, E7	Plasma	Detection, prognosis, Post-treatment surveillance	Retrospective	0.77 (combined)	1.00 (combined)	HPV: 24:6 non-HPV: 17:3	HPV: 47.3–91.7 non-HPV: 46.5–75.2	HPV: 64.9 non-HPV: 59.9	Tobacco, alcohol, tumor stage, treatment received
Chera et al. [14]. 2020	HPV ctDNA	Plasma	Post-treatment surveillance	Prospective	1.00 (two consecutive abnormal levels)	0.99 (two consecutive abnormal levels)	101:14	33–84	59	Tobacco

* Data was not found in the article.

**Table 2 ijms-23-14336-t002:** Summary of RNA biomarkers for HPV-positive OPC.

Study	Biomarker	Biofluid	Application	Study Type	Sensitivity	Specificity	No. of Patients (M/F)	Age Range	Mean Age	Other Covariates Recorded
Chernesky et al. [16]. 2018	E6/E7 mRNA	Saliva; oral swabs; FNA	Detection	Cross-sectional	/	/	50:9	40–80	59.8	/
Oliva et al. [17]. 2021	rRNA	Saliva	Post-treatment	Prospective	/	/	19:3	50–71	61	Tobacco, tumor stage

**Table 3 ijms-23-14336-t003:** Summary of epigenetic biomarkers for HPV-positive OPC.

Study	Biomarker	Biofluid	Application	Study Type	Sensitivity	Specificity	No. of Patients (M/F)	Age Range	Mean Age	Other Covariates Recorded
Misawa et al. [18]. 2020	Methylated genes	Plasma	Prognosis; post-treatment surveillance	Retrospective	/	/	217:35	32–95	65	Tobacco, alcohol
Shen Et al [19]. 2020	Methylated genes	Saliva	Post-treatment surveillance	Retrospective	0.90 (combined panel of HR E5L2-4 and EDNRB)	0.81 (combined panel of HR E5L2-4 and EDNRB)	62:8	/	60.2	Tobacco, alcohol

**Table 4 ijms-23-14336-t004:** Summary of EV biomarkers for HPV-positive OPC.

Study	Biomarker	Biofluid	Application	Study Type	Sensitivity	Specificity	No. of Patients (M/F)	Age Range	Mean Age	Other Covariates Recorded
Peacock et al. [30]. 2018	Micro-RNA (Details listed in Appendix A)	EVs	Detection	Retrospective	/	/	/	/	/	/
Mayne et al. [31]. 2020	Micro-RNA (Details listed in Appendix B)	EVs	Recurrence	Cross-validation	90%	79%	36:3	47–74	58	Tobacco

**Table 5 ijms-23-14336-t005:** Summary of oncoprotein biomarkers for HPV-positive OPC.

Study	Biomarker	Biofluid	Application	Study Type	Sensitivity	Specificity	No. of Patients (M/F)	Age Range	Mean Age	Other Covariates Recorded
Dickinsoni et al. [32]. 2020	Oncoprotein	Serum	Classification	Retrospective	/	/	22:14	36.6–84.7	62.6	/
Chernesky et al. [16]. 2018	HPV E6 oncoprotein	Saliva, FNA	Detection	Cross-sectional	/	/	50:9	40–80	59.8	/
Economopoulou et al. [33]. 2019	HPV16 E6/E7 CTCs	Blood	Prognosis	Prospective	/	/	22	41–76	48	Tobacco, alcohol, tumor stage, treatment received

**Table 6 ijms-23-14336-t006:** Summary of immune response-related biomarkers for HPV-positive OPC.

Study	Biomarker	Biofluid	Application	Study Type	Sensitivity	Specificity	No. of Patients (M/F)	Age Range	Mean Age	Other Covariates Recorded
Masterson et al. [34]. 2016	CD4^+^; CD8^+^; T-cells	Blood	Prognosis	Prospective	/	/	41:10	/	58	Tobacco, alcohol
Witzleben et al. [35]. 2021	Antibodies to HPV DNA E2 E7	Serum	Prognosis, treatment	Cross-sectional	/	/	67:10	35–77	56	Tumor stage, treatment received

**Table 7 ijms-23-14336-t007:** Summary of genetic biomarkers for HPV-negative OPC.

Study	Biomarker	Biofluid	Application	Study Type	Sensitivity	Specificity	No. of Patients (M/F)	Age Range	Mean Age	Other Covariates Recorded
Puttipanyalears Et al [36]. 2018	TRH gene methylation	Saliva	Diagnostic; no prognostic value	Bioinformatics with validation	82.61%	92.59%	24	/	/	/
Wise-Draper et al [37]. 2018	DEK gene	Plasma	Prognosis	Retrospective	/	/	30:6 *	/	56.67	Race distribution, smoking, alcohol
Burcher et al [38]. 2021	DDR gene mutation	Blood	Treatment	Retrospective	/	/	123:47 *	/	60	ETOH status, smoking, tumor stage, HPV status

* including patients from other HNSCC sites.

**Table 8 ijms-23-14336-t008:** Summary of protein biomarkers for HPV-negative OPC.

Study	Biomarker	Biofluid	Application	Study Type	Sensitivity	Specificity	No. of Patients	Age Range	Mean Age	Other Covariates Recorded
Boldrup [39] et al. 2017	NT-3	Plasma	Diagnosis	Retrospective	/	/	19	/	/	/
Sanada et al. [40]. 2020	LOXL2	Serum	Diagnosis	Retrospective	/	/	31:5 (M:F)	/	66:59	Tumor stage
Chang et al. [41]. 2020	MMP-1	Saliva	Diagnosis	Retrospective	70.37%	74.64%	27	/	/	Smoking
Lassig et al. [42]. 2017	MMP-1; MMP-3	Surgical drain fluid	Prognosis	Prospective	/	/	15:5	/	63.5	Smoking, alcohol
Carpén et al. [43]. 2019	TIMP-	Serum	Prognosis	Retrospective	/	/	66:24	36.6–84.	61.8	Smoking, alcohol, tumor stage

**Table 9 ijms-23-14336-t009:** Summary of immune response-related biomarkers.

Study	Biomarker	Biofluid	Application	Study Type	Sensitivity, Specificity	No. of Patients (M/F)	Age Range	Mean Age	Other Covariates Recorded
Mohajertehran et al. [53]. 2019	LDH	Saliva	Early diagnosis	Retrospective	/	33:11	27–83	59.61	/ *
Boldrup et al. [54]. 2021	IL1Ra	Blood	Diagnosis (failed), staging	Retrospective	/	87	19–84	/	Tobacco, fasting status
Poropatich et al. [55]. 2017	CTLA-4^+^ PD-1^+^	Blood	Staging	Retrospective	/	19:7	/	61	Tobacco, alcohol
	CD8^+^ T-cells								
Ida et al. [56]. 2021	T_RM_-like cells	Blood	Failed to correlate with staging	Retrospective	/	54:6	39–89	65	HPV status
Wilms et al. [57]. 2020	PD-L1 protein	Blood	Staging & prognosis	Retrospective	/	50:51	19–89	62	Tobacco, alcohol, treatment received
Zheng et al. [58]. 2021	CD73	Blood	Staging & prognosis	Retrospective	/	42:11	28–78	57.3	/
Gorphe et al. [59]. 2018	Anaemia;	Blood	Prognosis	Retrospective	/	126:41	38–77	59.2	HPV-status, smoking history (30 pack years), tumor stage
	Neutrophil-lymphocyte ratio								
Wichmann et al. [60]. 2017	HLA traits in leukocytes	Blood	Prognosis	Cross-sectional	/	78:12	/	/	Tobacco, alcohol, HPV status, treatment
Al-Samadi et al. [61]. 2017	Mononuclear cells	Blood	Prognosis (failed correlation)	Prospective animal model	/	/	/	/	/
Lassig et al. [42]. 2017	MMP-1, MMP-3; sFlt-1	Post-surgical drain fluid	Prognosis	Prospective	/	15:5	/	63.5	Tobacco, alcohol

* Data was not found in the article

## Data Availability

Not applicable.

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
