# Peer review of "Fluid Biomarkers in HPV and Non-HPV Related Oropharyngeal Carcinomas: From Diagnosis and Monitoring to Prognostication—A Systematic Review"

_ijms, 2022, doi:10.3390/ijms232214336_

Round 1

Reviewer 1 Report

Dear Authors

I have reviewed your paper with great interest.

I will accept your paper after a minimal revision.

My revision is:

Title: Very Good

Abstract: Very Good

Introduction and AIM: The problem and the aim are well descripting.

Materials, Patients and methods and statistics: All good.

Results: Focus on and well described.

Discussion and Thread: effectiveness Focus ON.

Tumor necrosis factor alpha secreted from oral  carcinoma correlated with or without HPV, it contributes to cancer pain and associated inflammation, may it reduce with some therapy?

Cite and discuss this paper:

Ripani U, Bisaccia M, Meccariello L. Dexamethasone and Nutraceutical Therapy Can Reduce the Myalgia Due to COVID-19 - a Systemic Review of the Active Substances that Can Reduce the Expression of Interlukin-6. Med Arch. 2022 Feb;76(1):66-71. doi: 10.5455/medarh.2022.76.66-71. 

References: Well chosen but to improve

Figures and Table: Very Good.

Author Response

Dear Reviewer,

Thank you for the comments.

We have adjusted our title and added 'Systematic Review'.

We have also trimmed down our abstract and made some changes to the introduction and conclusion, as well as adding more discussion on future perspectives.

Regarding methodology, we have reviewed our PRISMA flowchart, corrected errors, and added inclusion and exclusion criteria.

We have corrected some minor spacing issues in the results section as well as expanding some abbreviations.

Further elaboration has also been added to the discussion section.

We have spotted an error in referencing and have revised accordingly.

Regarding figures and tables, we have added references to each article in the tables, added sensitivity specificity values, and have also added a new figure 2. summarizing the fluid biomarkers by their location and HPV status.

Many thanks once again for the comments!

Reviewer 2 Report

Lee et al. present a comprehensive review that highlights the current state-of-art fluid biomarkers detection of oropharyngeal carcinoma. While the review is extensive, addressing the following points can sharpen the review:

Major points:

1.     In Figure 1, the flow diagram needs to be edited for clarity. The arrow starting from records screened to reports sought for retrieval should actually start from reports excluded. Please also elaborate on how the reports assessed for eligibility came down to 47?

2.     In Figure 1, 7 reports have been shown to excluded from 47 eligible reports. However, only 38 reports were included in the review leaving two reports unaccounted for. Please explain.

3.     In line 77, how were the tiles/abstracts deemed as eligible for inclusion? Details could also be included in Figure 1.

4.     In all the tables, add specificity and sensitivity values where applicable. Maintain uniform units for these values.

5.     In table 4, list out the miRNAs in the biomarker column.

6.     Adding a visual representation summarizing the type of biomarkers discussed in the study can be impactful.

7.     Several repeated sentences in conclusion and future perspective section and the abstract section need to be removed. Elaborate and provide justification for all the statements made in the abstract from lines 21-30 on the future scope of fluid biomarkers in prognosis and for potential treatment options in the conclusion and future perspectives section.

Minor Points:

8.     Italicize words in vitro, in vivo throughout the manuscript.

9.     In line 18, expand HPV.

10.  In all the tables, mention the year of the article and reference number in the study column for all the specified studies.

11.  In table 1, there needs to be : after non-HPV in age range column.

12.  In table 1, number of patients (M/F) are presented as a decimal value or ratios. Maintain consistent representation.

13.  In line 138, please elaborate on what area under the curve represents.

14.  In line 149-152, elaborate on the AJCC staging system and what T,N,M represent.

15.  All the table titles should be consistent to end with a period.

16.  In line 246, expand FNA.

17.  In line 269, it should be exploration.

18.  In line 381, expand PFS and OS.

19.  In line 394, expand DFS.

20.  In line 513, space is needed after 2.78.

Author Response

Dear Reviewer,

Thank you for the detailed comments and feedback, we have made several changes accordingly.

Points 1,2,3: Thank you for spotting the errors, we have recounted the number of articles and corrected the numerical errors in the flowchart as well as the methodology section. We have also added inclusion and exclusion criteria, both in Figure 1 (flowchart) and the methodology section.

4. Thank you for the suggestion, the values have been added where available and appropriate.

5. The microRNA biomarkers were originally not mentioned in the table due to the sheer number (33 in one article). Our team is currently working on how we can present the biomarkers in a clear and comfortable way in the tables.

6. Absolutely. We have included a summary figure on all the fluid biomarkers identified in this paper, grouped according to the type of biofluid and HPV-status. (Figure 2)

7. The abstract has been trimmed down; The numerical errors in the introduction and methodology sections has been revised; Further elaboration has been added to the discussion & future perspectives section.

Minor points:

Points 8-15, 19-20: Thank you for the detailed feedback, we have all revised these errors accordingly.

Points 16 & 18: We have already expanded FNA, PFS, OS in previous paragraphs.

Once again, thank you for the constructive and helpful feedback and comments!